Short-term interval aerobic exercise training does not improve memory functioning in relapsing-remitting multiple sclerosis—a randomized controlled trial

Baquet Lisa 1 l.baquet@web.de
Hasselmann Helge 2
Patra Stefan 3
Stellmann Jan-Patrick 1 4
http://orcid.org/0000-0002-3737-6402 Vettorazzi Eik 5
http://orcid.org/0000-0003-4899-8466 Engel Andreas K. 6
Rosenkranz Sina Cathérine 1
Poettgen Jana 1 4
Gold Stefan Michael 1 2
http://orcid.org/0000-0002-2411-6732 Schulz Karl-Heinz 3
http://orcid.org/0000-0001-8131-9467 Heesen Christoph 1 4 heesen@uke.de
1 Institute for Neuroimmunology and Multiple Sclerosis (INIMS), University Medical Center Hamburg-Eppendorf , Hamburg , Germany
2 Department for Psychiatry and Psychotherapy, Charité—Universitätsmedizin Berlin, Campus Benjamin Franklin , Berlin , Germany
3 Universitäres Kompetenzzentrum für Sport- und Bewegungsmedizin (Athleticum) und Institut und Poliklinik für Medizinische Psychologie, University Medical Center Hamburg-Eppendorf , Hamburg , Germany
4 Department of Neurology, University Medical Center Hamburg-Eppendorf , Hamburg , Germany
5 Institute of Medical Biometry and Epidemiology, University Medical Center Hamburg-Eppendorf , Hamburg , Germany
6 Department of Neurophysiology and Pathophysiology, University Medical Center Hamburg-Eppendorf , Hamburg , Germany
Hung Tsung-Min
Electronic publication date: 2018 Dec 12
Publication date: 2018
Volume: 6
Electronic Location ID: e6037
Received 2018 Jun 20; Accepted 2018 Oct 30
Copyright: © 2018 Baquet et al.
Copyright year: 2018
Copyright holder: Baquet et al.
License: This is an open access article distributed under the terms of the Creative Commons Attribution License, which permits unrestricted use, distribution, reproduction and adaptation in any medium and for any purpose provided that it is properly attributed. For attribution, the original author(s), title, publication source (PeerJ) and either DOI or URL of the article must be cited.
License URL: https://creativecommons.org/licenses/by/4.0/

Keywords: Multiple sclerosis, Cognition, Memory function, Aerobic exercise training, Neuroimmunology, Neurorehabilitation

Funding: German Ministery of Research and Education within the “Biopharma Campaing” in the “Neu2” Consortium This work was funded by the German Ministery of Research and Education within the “Biopharma Campaing” in the “Neu2” Consortium. The funders had no role in study design, data collection and analysis, decision to publish, or preparation of the manuscript.

==============================
Background

Only few aerobic exercise intervention trials specifically targeting cognitive functioning have been performed in multiple sclerosis.

Objective and Methods

This randomized controlled trial with 34 patients in the intervention group (IG) (mean: 38.2 years (±9.6)) and 34 patients in the control group (CG) (mean: 39.6 years (±9.7)) aimed to determine the effects of aerobic exercise on cognition in relapsing-remitting multiple sclerosis (RRMS). The primary outcome was verbal learning assessed by the verbal learning and memory test (VLMT). Patients were randomized to an IG or a waitlist CG. Patients in the IG exercised according to an individually tailored training schedule (with two to three sessions per week for 12 weeks). The primary analysis was carried out using the intention-to-treat (ITT) sample with ANCOVA adjusting for baseline scores.

Results

A total of 77 patients with RRMS were screened and 68 participants randomized (CG n = 34; IG n = 34). The sample comprised 68% females, had a mean age of 39 years, a mean disease duration of 6.3 years, and a mean expanded disability status scale of 1.8. No significant effects were detected in the ITT analysis for the primary endpoint VLMT or any other cognitive measures. Moreover, no significant treatment effects were observed for quality of life, fatigue, or depressive symptoms.

Conclusion

This study failed to demonstrate beneficial effects of aerobic exercise on cognition in RRMS. The trial was prospectively registered at clinicaltrials.gov (NCT02005237).

Introduction

Multiple sclerosis (MS) is the most common neurological disease in young people leading to substantial disability (Reich, Lucchinetti & Calabresi, 2018). MS is regarded a presumably T-cell driven autoimmune disease and the pathological hallmarks are inflammation and demyelination of the central nervous system (CNS) with a variable extent of infiltrating lymphocytes and macrophages, as well as activated microglia (Reich, Lucchinetti & Calabresi, 2018). However, progressive neurodegeneration plays an important role in MS pathobiology and occurs early on in the disease process and may even be involved in triggering inflammatory responses (Stys et al., 2012). Disease modifying therapies (DMTs) can control the inflammation to a variable extent and are thus able to decrease the frequency of relapses (Wingerchuk & Weinshenker, 2016). However, there is only modest evidence that the long-term disease progression is altered by DMTs. Neurodegeneration is thought to be a major driver of persistent disability and MS-related impairments in higher order brain functions such as cognition, highlighting the importance of developing treatment options in this area (Amato et al., 2013).

Exercise training has received increasing attention as a putative strategy to target the degenerative component of MS (Motl et al., 2017). Some (Kim & Sung, 2017), albeit not all (Klaren et al., 2016) studies conducted in experimental autoimmune encephalomyelitis (EAE) (the animal model of MS) have indicated a neuroregenerative effect of exercise. For example, Kim & Sung (2017) showed that exercise can improve memory function and increase hippocampal neurogenesis in EAE.

Different exercise training strategies have been applied in MS and endurance and resistance training seem to both exert some beneficial effects (Latimer-Cheung et al., 2013). Interval training approaches have received high acceptance as more time-efficient and attractive to patients (Garber et al., 2011). In MS, avoiding overheating seems highly relevant as it may induce the so called Uhthoff phenomenon leading to nerve conduction disturbances. Even interval high intensity training has been shown to be safe and effective in patients with MS with already relevant disability (Campbell, Coulter & Paul, 2018). However, while numerous studies now have shown beneficial effects of exercise training on physical capacity, strength, quality of life, mood (Latimer-Cheung et al., 2013) and fatigue in MS (Heine et al., 2015), high quality clinical trials focusing on cognitive function are scarce and the evidence is not conclusive (Sandroff et al., 2016). To our knowledge, no study yet has addressed cognition as its primary endpoint.

Outside of MS research, an increasing body of clinical data from related fields support the neuroprotective potential of exercise. For example, in healthy elderly humans, aerobic exercise training selectively increased the size of the anterior hippocampus and change in hippocampal volume was significantly related to improved memory in the exercise group (Erickson et al., 2011). Epidemiological studies have reported an association of physical fitness with lower risk of dementia disorders (Winblad et al., 2016). In addition, lifestyle interventions including exercise and nutrition counselling can reduce the risk of cognitive deterioration in people with minimal cognitive deficits (Ngandu et al., 2015). Thus, cognitive performance seems to be a suitable endpoint to determine neuroprotective effects of exercise interventions.

In the current study, we aimed to determine the effects of individually tailored aerobic exercise on verbal learning and memory, a key cognitive disability domain in MS, as the primary endpoint in patients with relapsing-remitting multiple sclerosis (RRMS).

Materials and Methods

Study design, overview and patient recruitment

Our trial was a single-blinded, 1:1 randomized, controlled phase IIa study with an active intervention group (IG) (n = 34; mean age: 38.2 years (±9.6)) and a waitlist control group (CG) (n = 34; mean age: 39.6 years (±9.7)). Patients in the IG underwent bicycle ergometry training with two to three sessions per week for 12 weeks and a subsequent extension phase of another 12 weeks. During the extension phase, patients originally randomized to the waitlist group had the opportunity to train as well and IG patients were invited to continue training. Details regarding the intervention are provided below.

The primary endpoint of our study was the verbal learning and memory test (VLMT) (Lux, Helmstaedter & Elger, 1999). Secondary endpoints were other cognitive functions (see below for details), as well as neuroimaging parameters (which will be reported separately). Tertiary endpoints were walking ability and motor function as well as patient reported outcomes (for details see below).

All endpoints were obtained at the MS Day Hospital at the University Medical Center Hamburg Eppendorf at baseline (t0), after the intervention (week 12; t1) and at the end of the extension phase at week 24 (t2). Patient recruitment was conducted by screening the registry database of the MS Day Hospital at the University Medical Center Hamburg Eppendorf for patients who met inclusion criteria and had indicated an interest in information about new clinical studies. Recruitment started in January 2013 and was completed in November 2015.

Standard protocol approvals and patient consent

The trial was approved by the ethics committee of the Hamburg Chamber of Physicians, (Registration Number PV4356). Participants gave written informed consent before enrolling in the study. The study was prospectively registered at ClinicalTrials.gov (identifier NCT02005237).

Inclusion and exclusion criteria

To be eligible for trial participation, patients had to be diagnosed with RRMS according to the McDonald criteria 2010 (Polman et al., 2011), an expanded disability status scale (EDSS) score <3.5 (Kurtzke, 1983), and currently in remission with no relapse or progression during the last 3 months. Patients had to be on stable immunotherapy for more than 3 months or without any planned change in disease-modifying therapies for the next 6 months.

Exclusion criteria were severe cognitive impairment or major psychiatric comorbidity (e.g., severe depression, psychosis, dementia) based on clinical judgement. Patients not capable to undergo aerobic exercise for medical reasons (i.e., due to heart disease) were excluded.

Sample size, randomization and masking

Sample size calculation was based on the results obtained in our previous trial (Briken et al., 2014) using PASS 2008. For the VLMT learning subtest we calculated group sample sizes of 20 each to achieve 90% power to detect a difference of −5.8 points with a standard deviation of 5 points on the scale. For the VLMT delayed recall we calculated group sample sizes of 17 and 17 to achieve 92% power to detect a difference of −2.6. Since this trial had a dual primary endpoint (VLMT learning and VLMT delayed recall), significance level (alpha) for each was set as 0.025 using a two-sided two-sample t-test. For the dual primary endpoints, 90% of power could thus be achieved by recruiting n = 20 patients for each arm of the trial. To safeguard against dropouts and the possibility that the effect sizes in the pilot trial had overestimated the real effect, we planned to enroll at least n = 60 patients (approximately n = 30 in each trial arm). After reaching the intended sample size the trial was stopped.

Inclusion was performed by physicians from the MS Day Hospital. Using a computer-based random number generator, we performed randomization by concealed allocation. The random sequence was generated by trial statistician E. Vettorazzi. While blinding of the participants was not possible, all outcome measures were obtained by assessors blind to group assignment.

Intervention

The intervention comprised 12 weeks of aerobic exercise on a bicycle ergometer. The prespecified interval training schedule was tailored to the patients’ individual level of aerobic fitness as determined by a spiroergometric exhaustion test at baseline (see trial protocol for details). The aim was to deliver an easily bearable interval training at moderate intensity that would be tolerable and feasible to the patients to ensure optimal adherence. The patients started with an intensity that was based on 10 different parameters: the power at Karvonen 60%, at 60% of Pmax, at 60% VO2peak, at 70% Borg 16 (20-point Borg scale; Borg, 1982), at 75% LT2 (lactate threshold), at 75% VT2 (ventilatory threshold), at VT1, at LT1_1.5 mmol, at Borg 11 and at a RQ = 0.91 (respiratory quotient). A training schedule with predefined 120 training sessions was developed for each patient. Each session comprised three to five intervals with a duration 3–20 min each. Training intensities ranged from 100% in training session 1 to 310% in training session 120. According to the individual level of fitness, patients started exercising with higher or lower intensity training sessions (training session 20 demanded more physical condition than training session 1). The breaks between the intervals were defined as 2–5 min according to the patients individual perceived exertion. The Borg rating during an ongoing training session defined intensity of the following training. During the training patients used a 10-point Borg scale (Borg, 1982) while the 20-point Borg scale was used within the ergometries at the t0 and t1 assessment. If all intervals were rated with Borg 1, 2 or 3 (“light or extremely light”) the patients could skip three sessions. If all intervals were rated Borg 7 (“somewhat hard”) this session was repeated the next time. If all intervals were rated Borg 8 or higher the patients did a lower intensity session next time. The patients had to train two to three times a week for 12 weeks. Duration and power (in Watts) in each training session were gradually increased to achieve a continuous increase of performance while keeping the perceived (Borg scale) and objectively measured (heart rate) exhaustion at a constant level. Heart rate was continuously recorded during exercise. Details are provided in the trial protocol (see Supplementary Material). An experienced physiotherapist who was familiar with the training program supervised the training of a maximum of three patients at a time.

Outcome measures

Cognitive functions

Our cognitive test battery assessed key cognitive disability domains in MS. Most of the tests are part of the internationally validated Brief Cognitive Assessment in MS (Filser et al., 2018). Moreover, we added tests from the battery of attention as we believe attention is a key domain of daily functioning and our previous work has indicated sensitivity to exercise interventions too (Briken et al., 2014).

The primary endpoints were the learning and the delayed recall scores of the VLMT (Lux, Helmstaedter & Elger, 1999). Specifically, we used the learning curve (words memorized in trials one to five) and the delayed recall as a measure of memory. This choice was based on several considerations: (1) While processing speed is often considered the most sensitive screening test for cognitive impairment in MS (Costa et al., 2016), impairments in learning and memory are also highly common in this population (Chiaravalloti & DeLuca, 2008). Recently it has been emphasized again that processing speed and memory are the key domains being affected (Sumowski et al., 2018); (2) basic science and clinical trials in healthy aging have indicated that CNS networks involved in learning and memory are particularly affected by exercise (Loprinzi, Edwards & Frith, 2017) and (3) results from our previous trial in progressive MS suggested that verbal learning and memory was most strongly improved by exercise. For the VLMT 15 words are read out to the patient five times and after each turn the patient had to remember and repeat as many words as possible. The final score is calculated by the total number of correctly remembered words. After 30 min the patients were asked to repeat the words again without those being read out by the assessor (delayed recall) (Filser et al., 2018).

Other neuropsychological functions were tested as secondary endpoints. We chose the symbol-digit-modalities test (SDMT) in its oral version and the paced auditory serial addition test (PASAT) to assess information processing (Fischer et al., 1999) (López-Góngora, Querol & Escartín, 2015). In the SDMT patients have 90 s time to assign the numbers one to nine to symbols that were prior associated to a certain number (Filser et al., 2018). For the PASAT patients are presented numbers on a screen every 3 s and have to add the new number to the one before (Rosti-Otajärvi et al., 2008). Visuospatial learning and memory was assessed using the brief visuospatial memory test-revised (BVMT-R) (Tam & Schmitter-Edgecombe, 2013). For the BVMT-R patients were presented six geometrical figures for 10 s and had to memorize their exact location on a grid. Subsequently they had to draw the figures correctly in the right place. This procedure is repeated three times (Filser et al., 2018). In addition, we applied the Corsi block-tapping task (Berch, Krikorian & Huha, 1998) to assess short-term and working memory. The assessor taps wood blocks in a certain order and the patient has to repeat the tapping in the same sequence (Kessels et al., 2000). Three subtests of the test battery for attention were used to assess attention—alertness, covert shift of attention (CSA) and incompatibility. For the subtest “alertness” patients had to press a button when a cross appeared on the screen and the reaction time was measured (tonic alertness). This test was also repeated with a warning signal before the appearance of the cross (phasic alertness). For the subtest “CSA” patients are presented with visual signals either giving right or wrong hints to the place where the main signal might appear or without a hint to the place of the following signal. The reaction time to the detection of the signal is measured. The subtest “incompatibility” assesses processing of parallel stimulus information. Arrows pointing to the right or the left appear on the screen and patients have to press a button if the direction of the arrow and the screen side where it appears match (Zimmermann & Fimm, 1994). We administered the “regensburg verbal fluency test” as a measure of verbal fluency. Patients had to find as many words as possible starting with a certain letter (verbal fluency I), from a certain category (verbal fluency II) or from two different categories (verbal flexibility) in a limited time (Aschenbrenner, Tucha & Lange, 2000). To assess social cognition/theory of mind, we utilized an abbreviated version of the “movie for the assessment of social cognition” (Dziobek et al., 2006; Pöttgen et al., 2013).

Expanded disability status scale

The EDSS was obtained by an experienced neurologist as the gold standard MS disability measure (Kurtzke, 1983). EDSS scores range from 0 = no disability to 10 = death due to MS and are based on assessment of different functional neurological domains. This nonlinear scale is mainly influenced by walking abilities between scores 4 and 7.5. At the lower end results from clinical examination and at the upper end by self-care abilities are major determinants.

Motor function and walking ability

The six minute walking test (6MWT) (Goldman, Marrie & Cohen, 2008) and the timed 25-foot walk (T25FW) (Kaufman, Moyer & Norton, 2000) served to assess walking ability. For the 6MWT patients had to walk for 6 min while the distance they cover was measured (Leone et al., 2015). In the T25FW patients had to walk 25 foot as fast as possible but with a safe gait (Supratak et al., 2018). The nine-hole peg test (9HPT) assessed motor function of the upper body. Nine pegs have to be placed and removed from the holes as fast as possible but picking up only one at a time (Feys et al., 2017).

Patient-reported outcome measures

We included the 16-item inventory of depressive symptomatology, which is a self-administered questionnaire with 16 items to quantify depressive symptoms (Barry et al., 2018; Fischer et al., 2015; Rush et al., 1996). The fatigue scale for motor and cognitive function, a 20-item questionnaire assessing cognitive and motor fatigue, was administered as well (Penner et al., 2009). The hamburg quality of life questionnaire for multiple sclerosis, which quantifies symptom impact of fatigue/thinking, upper and lower limb mobility, social function and mood (Gold et al., 2001) was included as well as the multiple sclerosis walking scale-12, a 12-item questionnaire assessing MS-related mobility limitations (Hobart et al., 2003).

Aerobic fitness

Bicycle spiroergometry using a Metalyser 3b (CORTEX Biophysik GmbH, Leipzig, Germany) was performed at baseline, after the intervention (week 12), and at the end of the extension phase (week 24). Peak oxygen intake (VO2peak), oxygen intake per kilogram of body weight (VO2peak/kg) and maximal power (Pmax) as well as Borg scaling and lactate levels were obtained every 2 min. The test started with 10 W after a 2-min resting and a 2-min warm-up phase at 10 W. Performance was continuously increased by one W every 6 s which resulted in a ramp of 20 W/2 min. We obtained continuous recordings of stress ECG to monitor cardiovascular function. Patients were instructed to exercise until perceived exhaustion (Borg 18–20 on a 20-point Borg scale). Spiroergometry results at baseline provided the anchor for the individualized training schedule which was determined by an algorithm including several maximal and submaximal parameters of the spiroergometry session (Power, oxygen intake (VO2), Rating of Perceived Exertion Scale (Borg scaling), lactate thresholds, ventilatory threshold, Karvonen-Formula, Respiratory Exchange Rate). Training intervals were performed near the aerobic threshold level within an individually determined starting point to improve aerobic capacity (Westhoff et al., 2013). For training plan details see protocol in the Supplementary File.

Statistical analyses

The primary analysis compared changes from baseline to week 12 between the IG and the waitlist CG. According to guidelines for statistical analysis of clinical trials published by The European Agency for the Evaluation of Medicinal Products (CPMP/ICH/363/96 and CPMP/EWP/2863/99), we computed the primary statistical analysis for all outcomes using ANCOVA models adjusting for baseline measurements of the respective outcome variable to evaluate treatment effects (measured as change from baseline). No other covariates were included in this primary analysis. As recommended, this model did also not include treatment by covariate interactions. All primary analyses were conducted as intention-to-treat (ITT) including all patients who had received group allocation. Every effort was made to obtain week 12 and week 24 data from all participants (even if they dropped out of the exercise program). In case of missing data, primary ITT analyses were conducted using a last-observation-carried-forward approach. As sensitivity analyses, we computed the same ANCOVA models as per-protocol (PP, i.e., including only patients who completed the 12 weeks of training with at least 18 training sessions).

We performed a further sensitivity analysis of treatment responders compared to the CG. Responders included those 11 patients (upper tercile) who had shown the strongest increase of maximum power values (Pmax) from baseline to week 12 (Pmax_week12—Pmax_baseline).

Results

A total of 77 patients were interested in participating in our study. After screening for eligibility, 68 patients fulfilled the inclusion criteria (for patient attrition see Fig. 1). They were randomized to the IG or a CG. On average, patients in the IG received 19.5 training sessions and there was at least 1 day rest between sessions. Notably, the proportion of men was higher in the IG than in the CG (38% vs 26%) (see Table 1). A total of 58 patients had complete data at baseline- as well as the follow-up-testing 12 weeks later. For the ITT analyses, all patients randomized were included (i.e., 34 patients in each group). The PP analyses included all patients who had completed at least 18 training sessions. A total of 23 participants of the 34 participants of the IG reached this goal. A total of 11 participants did not complete 18 training sessions, mostly due to personal reasons (family issues, job, holiday). Four of those 11 participants did not attend all three visits due to pregnancy, an ill family member, relapse and nonappearance as can be seen in Fig. 1.

Figure 1 Participant flow chart.

Table 1 Clinical baseline characteristics.

	CG	IG_ITT	IG_PP	IG_RG	
Baseline n = 34	Baseline n = 34	Baseline n = 23	Baseline n = 11	
Age (years)	39.6 (9.7)	38.2 (9.6)	38.6 (9.9)	38.2 (9.6)	
Sex (m/f)	9/25	13/21	9/14	5/6	
Education (years)	12.1 (1.5)	12.3 (1.4)	12.7 (1.0)	12.1 (1.6)	
EDSS Baseline	1.8 (1.0)	1.7 (0.9)	1.6 (0.9)	1.4 (0.9)	
Disease duration—symptoms (years)	9.1 (7.7)	8.1 (5.7)	7.2 (6.2)	5.8 (6.7)	
Disease duration—diagnosis (years)	5.7 (6.3)	6.8 (5.5)	6.6 (6.0)	4.4 (6.6)	
Immunotherapy (%)	44.1	67.7	69.6	72.7	
Number of sessions		19.5 (10.0)	24.7 (6.5)	26.1 (7.0)	
Notes:

Data as: mean (standard deviation).

CG, control group; IG, intervention group; ITT, intention-to-treat; PP, per-protocol; RG, responder subgroup; EDSS, expanded disability status scale.

Overall, patients had low to moderate clinical disability (median EDSS = 1.5; range 0–3.5). Cognitive function overall was in the range of what would be expected in early RRMS and as a group only showed mild impairment (PASAT showed the highest rate of individual abnormal values with baseline findings below age and education adapted references in n = 5 IG and n = 4 CG patients).

Effects of exercise on cognitive function

No significant treatment effects were detected for the primary endpoints (VLMT learning, VLMT delayed recall) or any of the other neuropsychological tests included (see Table 2). This was seen in the ITT analyses as well as the sensitivity analyses (PP and responder analyses). Moreover, analysis of 12-week follow-up of all available data (29 out of 34 patients of the IG) did not indicate any change in cognitive measures.

Table 2 Cognitive outcomes.

	IG	CG	Mean between group-difference [95% CI]	f-value*	P-value*	Effect-size* partial eta sq	
Baseline n = 34	Week 12 n = 34	Baseline n = 34	Week 12 n = 34	
VLMT 1–5	58.7 (9.1)	61.0 (8.0)	59.1 (8.3)	60.6 (7.6)	−0.7 [−3.2–1.9]	0.26	0.61	<0.01	
VLMT 5–7	0.8 (1.4)	1.2 (2.5)	1.0 (1.7)	1.1 (1.6)	−0.3 [−1.2–0.6]	0.40	0.53	0.01	
SDMT (points)	58.9 (12.2)	59.4 (12.8)	58.2 (9.1)	60.5 (10.9)	1.8 [−0.9–4.5]	1.80	0.18	0.03	
BVMT-R total learning (points)	25.8 (5.7)	25.3 (5.7)	26.4 (6.2)	26.0 (5.5)	0.3 [−1.7–2.3]	0.11	0.74	<0.01	
BVMT-R recall (points)	9.9 (2.0)	10.1 (1.6)	10.0 (2.0)	9.8 (2.0)	−0.4 [−1.1–0.3]	1.39	0.24	0.02	
BVMT-R recognition hits (points)	5.8 (0.5)	5.9 (0.3)	5.9 (0.3)	5.9 (0.5)	0.0 [−0.2–0.2]	0.00	>0.99	<0.01	
BVMT-R false alarms (points)	0.0 (0.2)	0.1 (0.2)	0.1 (0.3)	0.0 (0.0)	−0.1 [−0.2–0.0]	3.60	0.06	0.05	
TAP tonic alertness (ms)	250.0 (40.6)	259.5 (46.9)	259.2 (42.9)	259.0 (34.4)	−7.8 [−19.4–3.7]	1.84	0.18	0.03	
TAP phasic alertness (ms)	254.9 (44.8)	254.0 (41.8)	256.9 (43.4)	254.4 (28.0)	−0.7 [−13.3–11.9]	0.01	0.91	<0.01	
TAP CSA valid (ms)	303.8 (54.6)	298.8 (54.3)	301.6 (44.4)	295.9 (41.5)	−1.3 [−17.1–14.4]	0.03	0.87	<0.01	
TAP CSA invalid (ms)	342.8 (62.6)	338.0 (57.7)	348.4 (58.6)	345.8 (62.6)	3.7 [−16.4–23.9]	0.14	0.71	<0.01	
TAP incompatibility (ms)	489.2 (62.9)	494.9 (63.6)	488.0 (76.9)	485.8 (69.7)	−8.3 [−30.6–13.9]	0.56	0.46	0.01	
RWT verbal fluency I (points)	23.3 (8.1)	17.7 (6.1)	24.3 (7.7)	18.8 (6.9)	0.6 [−2.0–3.2]	0.21	0.65	<0.01	
RWT verbal fluency II (points)	36.9 (10.1)	28.3 (9.1)	36.6 (9.6)	27.6 (7.3)	−0.6 [−3.6–2.5]	0.13	0.71	<0.01	
RWT verbal flexibilty (points)	23.4 (5.0)	20.7 (4.9)	21.7 (4.5)	19.9 (5.1)	0.0 [−2.2–2.2]	0.00	0.99	<0.01	
PASAT (points)	46.7 (10.7)	47.9 (14.3)	49.8 (8.8)	53.6 (8.4)	2.9 [−0.9–6.7]	2.34	0.13	0.03	
Corsi forward (points)	9.0 (2.2)	9.2 (2.3)	8.8 (1.9)	9.6 (1.9)	0.5 [−0.2–1.2]	1.75	0.19	0.03	
Corsi backward (points)	8.5 (1.9)	8.6 (1.7)	8.5 (1.9)	9.0 (1.8)	0.5 [−0.1–1.1]	2.46	0.12	0.04	
MASC	11.3 (2.2)	11.5 (2.2)	11.6 (2.3)	12.2 (1.8)	0.4 [−0.3–1.2]	1.42	0.24	0.02	
Notes:

Mean values of raw scores (standard deviation).

IG, intervention group; CG, control group; VLMT, verbal learning and memory test; SDMT, symbol digit modalities test; BVMT-R, brief visuospatial memory test-revised; TAP, tonic alertness, test battery for attention tonic alertness; TAP phasic alertness, test battery for attention phasic alertness; TAP CSA valid, test battery for attention covert shift of attention valid; TAP CSA invalid, test battery for attention covert shift of attention invalid; TAP incompatibility, test battery for attention incompatibility; RWT, regensburger verbal fluency test; PASAT, paced auditory serial addition test; MASC, movie for assessment of social cognition.

* ANCOVA.

Effects of exercise on motor function and walking ability

Similarly, ITT analyses of motor function was negative with no significant effects in any of the three measures included (6MWT, T25, 9HPT dominant hand). One test (9HPT nondominant hand) showed significant effects in favor of the CG (see Table 3).

Table 3 Motor, training and patient reported outcomes.

	IG	CG	Mean between group-difference [95% CI]	f-value*	P-value*	Effect-size* partial eta sq	
Baselinen = 34	Week 12 n = 34	Baseline n = 34	Week 12 n = 34	
Motor function and aerobic fitness	
6 MWT (m)	432.1 (105.7)	454.4 (104.1)	448.8 (79.8)	466.7 (84.2)	4.0 [−36.5–44.5]	0.04	0.85	<0.01	
9 HPT dominant (s)	19.5 (3.7)	19.6 (3.6)	19.1 (2.9)	18.6 (3.0)	−0.6 [−1.4–0.1]	2.95	0.09	0.04	
9 HPT non dominant (s)	19.6 (2.9)	19.9 (3.0)	19.8 (4.1)	19.0 (3.5)	−1.1 [−2.0 to −0.2]	5.63	0.02	0.08	
T25FW (s)	4.8 (0.8)	4.8 (0.8)	4.8 (0.8)	4.8 (0.8)	−0.1 [−0.4–0.2]	0.48	0.49	0.01	
V02peak (ml O2/min)	2,151.5 (669.5)	2,225.6 (743.4)	1,761.5 (421.3)	1,779.4 (427.4)	−51.4 [−165.2–62.5]	0.81	0.37	0.01	
V02peak/kg ((ml O2/min)/kg)	27.2 (7.9)	28.0 (8.8)	25.6 (5.5)	25.6 (5.4)	−0.9 [−2.5–0.6]	1.40	0.24	0.02	
Pmax (Watt)	160.4 (45.4)	176.3 (55.4)	139.5 (31.1)	139.6 (31.0)	−15.7 [−27.1 to −4.3]	7.57	0.01	0.11	
Patient-reported outcome measures	
IDS-16SR	6.4 (5.1)	5.9 (4.4)	6.1 (4.3)	6.3 (4.6)	0.5 [−0.8–1.9]	0.60	0.44	0.01	
FSMC	51.5 (22.1)	51.1 (22.6)	53.4 (21.6)	50.9 (21.4)	−1.9 [−6.9–3.1]	0.60	0.44	0.01	
MSWS-12	19.4 (10.4)	19.6 (10.2)	18.7 (10.7)	18.7 (9.9)	−0.3 [−2.1–1.6]	0.08	0.78	<0.01	
HAQUAMS	53.1 (16.9)	53.7 (17.8)	51.2 (18.7)	51.8 (14.7)	−0.4 [−4.5–3.7]	0.04	0.84	<0.01	
Notes:

Data as mean (standard deviation).

IG, intervention group; CG, control group; 6 MWT, six minute walking test; 9 HPT, nine-hole peg test; T25FW, timed 25-foot walk; Pmax, maximal power; IDS16-SR, 16-item version of inventory of depressive symptomatology self-rated; FSMC, fatigue scale for motor and cognitive functions; MSWS-12, 12-item MS walking scale; HAQUAMS, hamburger quality of life questionnaire in multiple sclerosis.

* ANCOVA.

Effects of exercise on fitness

At baseline, fitness indices were in the range of 140–160 W at Pmax and 25–27 ml (O2/min)/kg at V02peak, thus indicating a low level of fitness in all patients. Significant treatment effects favoring the IG were observed for Pmax in the ITT analysis (see Table 3). This was confirmed in the PP analysis. However, VO2peak and VO2/kg showed no significant treatment effects.

Effects of exercise on patient-reported outcomes

Finally, there were no significant treatment effects in any of the patient-reported outcome measures including quality of life, fatigue, mood or self-reported walking ability (see Table 3).

Discussion

Overall, our trial of standardized exercise over 12 weeks in RRMS patients compared to a waitlist CG failed to meet any of its clinical endpoints. Specifically, the study did not show effects on any cognitive measure and failed to produce significant changes in patient related or objective outcome measures for functioning in MS. These results thus cast doubt on the generalizability of recent trials that suggested the potential of exercise to improve cognitive function in progressive (Briken et al., 2014) or RRMS (Zimmer et al., 2017).

Compared to our previous study in progressive MS patients (Briken et al., 2014), RRMS patients in the current trial were substantially less disabled physically and cognitively when entering the trial, thereby potentially limiting room for improvements (ceiling effect). Mean EDSS values were below 2.0 which were the lowest among 26 recently reviewed exercise interventions in MS assessing cognition as an outcome (Sandroff et al., 2016). In addition, all mean baseline scores on cognitive function were within one SD of normative values. In contrast, at least 40% of unselected RRMS cohorts show cognitive deficits (Chiaravalloti & DeLuca, 2008). Interestingly from the 26 studies referred to above, only two case report studies selected patients based on cognitive deficits. Our study underlines that further studies on the cognitive effects of exercise treatments need to address patients presenting with relevant cognitive deficit. Another aspect to consider is the choice of verbal learning and memory as our primary endpoint. As described in the methods section above, this choice was based on several considerations. However, it could be argued that—given its high sensitivity for detecting cognitive impairment in MS—a test of processing speed (e.g., the SDMT or the PASAT) might have been a more obvious choice. However, given our null effects across all cognitive domains tested as primary or secondary endpoints (including the SDMT and PASAT), this would not have changed our results.

One strength of our study is that cognition was predefined as the primary outcome and verbal learning and memory are key domains. Consistent with our earlier study in progressive MS, a recent trial on high intensity training could show effects on different cognitive dimensions with strongest effects in the verbal memory dimension of the Brief Cognitive Assessment in MS battery (Zimmer et al., 2017). However, in the few other MS exercise studies most effects were reported in information processing speed and executive functioning (Sandroff et al., 2016). From studies on healthy aging it appears that exercise may exert differential effects on various cognitive domains depending on age and intervention type (Hötting & Röder, 2013). For example, Hötting et al. (2012) have shown an improvement of episodic memory in middle-aged adults after cardiovascular training but enhanced attention scores after a combined stretching/coordination training. Thus, further work is needed to address which domains might be most sensitive for which intervention at which intensity and at which disease stage in MS.

Training intensity—albeit carefully tailored to the individual patient’s level of fitness at baseline—was only moderate and training frequency deliberately set at the lower end. Certainly, the low intensity may account for the negative findings. While available meta-analyses in MS indicate a consistent benefit of exercise training on fitness (Latimer-Cheung et al., 2013), this effect can be subtle or not detectable in individual trials. In our previous trial in progressive MS (Briken et al., 2014) with a similar number of training sessions over 10 weeks, VO2peak increases in the IG were small and between group comparisons reached significance mainly due to worsening of the CG. An earlier study in a mixed sample of relapsing and progressive patients of 8 weeks duration failed to detect significant increases in VO2peak (Schulz et al., 2004). Therefore, a more challenging training plan in terms of intensity and frequency might be considered in future trials, especially in patients with minimal neurological impairment. This will need to be weighed against potential disadvantages as adherence could become a problem with more intensive training programs, especially when also expanding the training period. Although our patients only had minor neurological disability, several patients had difficulties managing the training sessions at the center mainly due to other private obligations. More individual tailoring of the interventions such as at-home training with web-based supervision might be an approach here.

Effects of exercise training on cognitive functioning in other populations seem not to directly depend on fitness parameters. Resistance training or coordinative training regimen have been shown to improve different cognitive measures (Hötting & Röder, 2013). On the other hand, meta-analytic data in aging adults indicate that more training and higher training intensity is better (Gomes-Osman et al., 2018). Therefore, cardiovascular fitness indicators can only be regarded as surrogate markers and neither necessary nor sufficient outcomes for beneficial effects on brain functioning.

In a related matter, training duration might have been too short. MS exercise studies to date typically employ training programs of approximately 12 week duration (Latimer-Cheung et al., 2013) with the longest studies up to 26 weeks. Studies in aging healthy adults performed training up to 52 weeks and meta-analytic data indicate a relevant effect of intervention length (Northey et al., 2017). Intriguingly, the recent study by Zimmer indicated that high intensity training over just 3 weeks can have beneficial effects on cognition as well as on other functional domains in MS (Zimmer et al., 2017). Therefore, an optimal balance of intensity and duration needs to be achieved and weighted against other factors such as treatment adherence and attrition. In this context, acceptance of a control condition is a relevant issue and longer studies may need active control interventions to keep patients on study.

Finally, on a more optimistic note, it is conceivable that CNS effects of exercise occur on a subclinical level, that is, without detectable effects on formal cognitive testing. For example, we recently showed that resistance training can alter structural MRI markers in MS (Kjølhede et al., 2017). Thus, MRI might be a more suitable tool to detect early alterations of functional and structural brain status, particularly in short trials. In the current trial, we have obtained both measures of structural and functional connectivity, which will be reported separately.

Conclusion

In conclusion, we could not demonstrate a beneficial effect of a 12-week moderate exercise training in minor disabled patients with MS on a set of cognitive outcome measures. This negative finding can help to design further exercise trials in MS.

Supplemental Information

Supplemental Information 1 Raw data.

Click here for additional data file.

Supplemental Information 2 CONSORT Checklist.

Click here for additional data file.

Supplemental Information 3 AERCONN Protocol.

Click here for additional data file.

Supplemental Information 4 Cognitive outcomes–per-protocol (PP) analysis.

mean values of raw scores (standard deviation).

IG: Intervention group; CG: Control group.

VLMT: Verbal Learning and Memory Test; SDMT: Symbol Digit Modalities Test; BVMT-R: Brief Visuospatial Memory Test-Revised; TAP Tonic Alertness: Test Battery for Attention Tonic Alertness; TAP Phasic Alertness: Test Battery for Attention Phasic Alertness; TAP CSA valid: Test Battery for Attention covert shift of attention valid; TAP CSA invalid: Test Battery for Attention covert shift of attention invalid; TAP Incompatibility: Test Battery for Attention Incompatibility; RWT: Regensburger Verbal Fluency Test; PASAT: Paced Auditory Serial Addition Test; MASC: Movie for Assessment of Social Cognition; *ANCOVA.

Click here for additional data file.

Supplemental Information 5 Motor, training and patient reported outcomes–per-protocol (PP) analysis.

Data as mean (standard deviation).

IG: Intervention group; CG: Control group.

6 MWT: Six minute walking test; 9 HPT: Nine-Hole Peg Test; T25FW: Timed 25-foot walk; Pmax: maximal Power; IDS16-SR: 16-item version of Inventory of Depressive Symptomatology Self-Rated; FSMC: Fatigue Scale for Motor and Cognitive Functions; MSWS-12: 12-item MS Walking Scale; HAQUAMS: Hamburger Quality of Life Questionnaire in Multiple Sclerosis; *ANCOVA.

Click here for additional data file.

Supplemental Information 6 Cognitive outcomes–Responder (RG) analysis.

mean values of raw scores (standard deviation).

IG: Intervention group; CG: Control group.

VLMT: Verbal Learning and Memory Test; SDMT: Symbol Digit Modalities Test; BVMT-R: Brief Visuospatial Memory Test-Revised; TAP Tonic Alertness: Test Battery for Attention Tonic Alertness; TAP Phasic Alertness: Test Battery for Attention Phasic Alertness; TAP CSA valid: Test Battery for Attention covert shift of attention valid; TAP CSA invalid: Test Battery for Attention covert shift of attention invalid; TAP Incompatibility: Test Battery for Attention Incompatibility; RWT: Regensburger Verbal Fluency Test; PASAT: Paced Auditory Serial Addition Test; MASC: Movie for Assessment of Social Cognition; *ANCOVA.

Click here for additional data file.

Supplemental Information 7 Motor, training and patient reported outcomes–Responder (RG) analysis.

Data as mean (standard deviation).

IG: Intervention group; CG: Control group.

6 MWT: Six minute walking test; 9 HPT: Nine-Hole Peg Test; T25FW: Timed 25-foot walk; Pmax: maximal Power; IDS16-SR: 16-item version of Inventory of Depressive Symptomatology Self-Rated; FSMC: Fatigue Scale for Motor and Cognitive Functions; MSWS-12: 12-item MS Walking Scale; HAQUAMS: Hamburger Quality of Life Questionnaire in Multiple Sclerosis; *ANCOVA.

Click here for additional data file.

Additional Information and Declarations

Competing Interests

Author Contributions

Clinical Trial Ethics

Data Availability

Clinical Trial Registration

Jan-Patrick Stellmann received research grants and speaker honoraries from Biogen, Genzyme and Merck. Stefan Michael Gold has received research grants and honoraria from Biogen, Almirall and Mylan. Christoph Heesen received research grants and speaker honoraries from Biogen, Genzyme, Novartis and Merck. Helge Hasselmann, Stefan Patra, Eik Vettorazzi, Andreas K. Engel, Sina Cathérine Rosenkranz, Jana Poettgen, Karl-Heinz Schulz have no competing interests.

Lisa Baquet performed the experiments, analyzed the data, prepared figures and/or tables, authored or reviewed drafts of the paper, approved the final draft.

Helge Hasselmann analyzed the data, approved the final draft.

Stefan Patra performed the experiments, approved the final draft.

Jan-Patrick Stellmann authored or reviewed drafts of the paper, approved the final draft.

Eik Vettorazzi analyzed the data, approved the final draft.

Andreas K. Engel conceived and designed the experiments, approved the final draft.

Sina Cathérine Rosenkranz authored or reviewed drafts of the paper, approved the final draft.

Jana Poettgen authored or reviewed drafts of the paper, approved the final draft.

Stefan Michael Gold conceived and designed the experiments, analyzed the data, authored or reviewed drafts of the paper, approved the final draft.

Karl-Heinz Schulz conceived and designed the experiments, contributed reagents/materials/analysis tools, authored or reviewed drafts of the paper, approved the final draft.

Christoph Heesen conceived and designed the experiments, authored or reviewed drafts of the paper, approved the final draft.

The following information was supplied relating to ethical approvals (i.e., approving body and any reference numbers):

The trial was approved by the ethics committee of the Hamburg Chamber of Physician.

The following information was supplied regarding data availability:

The raw data is available as a Supplemental File.

The following information was supplied regarding Clinical Trial registration:

NCT02005237 (https://clinicaltrials.gov/ct2/show/NCT02005237?term=NCT02005237&rank=1).

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
