# Peer review of "Short-term interval aerobic exercise training does not improve memory functioning in relapsing-remitting multiple sclerosis—a randomized controlled trial"

_PeerJ, doi:10.7717/peerj.6037_

## Round 0.1 · original submission · Major Revisions

I have received two reviewers' comments. Although both reviewers expressed their interest in your study, several aspects of this manuscript should be revised to improve its clarity.

I regard the reviewers' opinions as substantive and well-informed. I believe that all of their comments require contemplation and appropriate attention in revising the document if it is to contribute appropriately to PeerJ and the extant literature. Please revise or refute according to the two reviewers' comments and provide a point by point reply in addition to the revised manuscript.

Tsung-Min Hung, PhD., FNAK
PeerJ editor
Research chair professor,
Department of Physical Education,
National Taiwan Normal University

Reviewer 1 ·

Basic reporting

Major issues-
Introduction-
1. The Intro should be reorganized and broadened. The authors may state the hallmarks of pathology of MS with one paragraph, followed by the potential role of exercise in attenuating cognitive impairments in RRMS (or MS), then state why you chose interval exercise, not continuous aerobic exercise, as an intervention.
2. Please specify why you chose to focus on memory in your manuscript. Accumulating evidences have supported that memory impairment is one of the core deficit in MS.
Methods-
1. Line 96-97: If the authors recruited participants from only one hospital, how the effect of selection bias mitigated in this study?
2. Line 112: how the authors quantify ‘severe’ cognitive impairment? Did the participants tested with their general cognitive ability, education level, socioeconomic status, and BMI (body fat %)? These factors are all closely related to cognitive function (or impairment).
3. Intervention: More details are needed with regards to the content of intervention. For example, when you implemented interval training, please specify how long was the between-set recovery? The exercise:rest ratio is crucial to the effectiveness of interval exercise training, therefore should be clearly stated in the manuscript. Also, please specify the intensity of the exercise (e.g., % maximal aerobic power). Please also specify: How many participants administered the intervention at the same time? How many participants wore a HR monitor in each session? How many physiotherapists supervised each session?
4. Did participants monitored and controlled regarding to their daily activities, such as reading and/or physical activity?
5. More details should be provided regarding outcome measures, including cognitive functions, motor function & walking ability, and patient-reported outcome measures. Please also justify your decision of task/assessment selection. Were the cognitive tasks or fitness assessments you used valid in MS? Currently, the format is too rough.
Results
1. Please provide statistical results, including mean and SD, p value, effect size, 95% CI) in the sections pertaining to effects of exercise on cognition, motor function and walking ability, fitness, and patient-reported outcomes.
Discussion
1. The reviewer believes that several potential limitations should be subjected to the current study. For example, as mentioned in previous comment, the authors should elaborate the potential confounding effects from several background variables, such as general cognitive ability, education level, socioeconomic status, and BMI, and how the confounding effects can be mitigated in the current study. Also, did selection bias exist in the current study? If yes, please elaborate this issue (and how it can be mitigated).

Specific comments-
1. Title- The reviewer suggests the authors to revise the term ‘aerobic exercise’. A more precise term such as ‘interval aerobic exercise training’ could be more appropriate as the term ‘aerobic exercise’ could refer to both continuous and interval exercise.
2. Line 26, 34, and 36: Please give the full name for ‘MS’, ‘RRMS, and ‘EDSS’ for first appearances.
3. Line 27: Please at least report the number of participants in each group and their mean and SD of age here.
4. Line 83-84: Likewise, please at least provide the number of participants as well as mean and SD of age in the IG and CG, respectively.
5. Line 108: Please specify what EDSS is for first appearance. It is possible that some readers are not familiar with the purpose and interpretation of this assessment.
6. Line 110: Did the authors collect data on how long they have had taken the immunotherapy? While some participants may have longer history of immunotherapy than others, would this inter-individual difference affect your results?
7. Line 114: Were participants’ readiness to physical activity confirmed by valid assessment, such as the Physical Activity Readiness Questionnaire (PARQ)?
8. Line 117: What software was used to calculate sample size? G*Power?
9. Line 121: If a two-tailed two-sample (independent) t-test was used for sample size calculation, please specify what metric (e.g., Cohen’s d) was adopted for detecting group difference.
10. Line 121: Why the authors chose an alpha of .025 and power of .90, rather than .05 and .80? This should be clarified.
11. Line 118-121: the authors stated several figures for detecting group difference (e.g., -5.8 pts on the VLMT learning subtest, -2.6 pts on the VLMT delayed recall test). Please justify the use of these figures or cite papers to support your decision.
12. Line 132: please justify the choice of a 12-week intervention.
13. Line 197-198: Please clearly elaborate the ANCOVA (e.g., 2 (group: IG, CG) x 3 (time: T0, T1, T2)).
14. Results- Did participants included in the ITT and PP differed in terms of baseline data??
15. Line 269: Should be ‘…less disabled, physically and cognitively, thereby…’
16. Line 274: the sentence is incomplete here.
17. Line 320: the sentence here is unclear to the reviewer.
18. Table 1, 2, and 3: The tables should be refined. For instance, please make sure all the value presented aligned to either the left or right side of each column. Also, the reviewer cannot find the T2 data in either Table 2 or Table 3.

Experimental design

no commment

Validity of the findings

no comment

Additional comments

The present study addresses a relevant topic and has the potential to contribute to the literature and treatment of multiple sclerosis. Novelties of the current manuscript include the use of interval training, the employment of a comprehensive measure on multiple cognitive domains that are vulnerable to the progression of multiple sclerosis, and the follow-up assessments. Key findings of the current study are that, in individual with relapsing-remitting multiple sclerosis (RRMS), 8 weeks of interval training has no benefit to cognitive performance, including memory, executive function, and processing speed. Generally speaking, the manuscript is well-written, and the study was conducted in a professional manner. The authors should be credited by their efforts. Reservation, however, should be given to several parts of the study. The reviewer’s major concerns are to the methodological ends, with several issues worth clarification, justification, or contemplation. Please find the comments attached.

Annotated reviews are not available for download in order to protect the identity of reviewers who chose to remain anonymous.

·

Basic reporting

Overall the paper is well-written and tables and raw data have been provided.

Experimental design

This study reports the results an RCT involving 69 people with RRMS examining the effects of AE on memory. The topic is a timely one and even though these are ‘negative’ findings’, the report of the design and results are important for future studies. Participants in the AE group trained on bicycle ergometry 2-3 times per week for 12 weeks. Importantly, outcomes were examined at 12 week follow-up which has been a weakness of previous trials.

Validity of the findings

Good RCT practices were followed, for example the trial was registered and the authors ensured random allocation and blinding of assessors. The authors included multiple outcomes making the results more convincing.

Additional comments

For the most part, the limitations and design flaws are addressed adequately. For example, the study was aimed at cognition but the participants were not asked whether they in fact had cognitive problems. It was not surprising then that there was a ceiling effect. This issue was mentioned in the Discussion section.
I have two major issues involving the intensity and duration of the AE intervention that participants received.
First of all, the maximum sessions that participants could have received was 30 but in fact participants attended on average 19.5 sessions (if 3X week this would be about 4 weeks). The intervention was 2-3 times per week but which was it- 2 or 3? The difference between 24 sessions and 30 session is substantial. The authors should discuss the appropriate length of treatment to gain an effect of AE gleaned from other studies in healthy individuals and in people with neurological disorders such as stroke. I would suggest based on the literature, that 8 weeks duration or about 24 sessions would be optimal. The issue of suboptimal treatment duration is not adequately discussed. Please compare and contrast short duration (<8weeks) with longer duration (>12 weeks) in order to provide context for these negative findings. There are multiple studies examining the effects of AE on cognition in stroke for example (Ploughman and Kelly 2016 PMID:27661010) In Fig 1 consort diagram, although not specifically indicated, it seems that 11 people were excluded from PP analysis because they attended <18 sessions. This is not quite clear-please indicate how many people were excluded due to this reason. Furthermore, why did these people not attend? Non-compliance issues are important for the planning of future studies. Please indicate reasons.
The second issue with the intervention is the low intensity nature. In the AERCONN protocol the exercise is anchored by W recorded at Borg RPE 11/20. This seem to be very low; Borg 11/20 is considered ‘light’ (https://www.cdc.gov/physicalactivity/basics/measuring/exertion.htm) and current guidelines support that training should be in the ‘somewhat hard’ category to achieve moderate level (according to the MS guidelines http://www.csep.ca/CMFiles/Guidelines/specialpops/CSEP_MS_PAGuidelines_adults_en.pdf). According to the AERCONN protocol, the first 20 sessions (the majority obtained by most people), the training was in the ‘light’ or ‘mild’ training threshold. This is a major issue that has not been adequately addressed in the Discussion section. Please review the intensity of exercise required to change cognition in animal models and clinical trials in order to place the work in context. Furthermore, the intensity of the intervention should be clearly indicated in the Methods section.
The title and abstract should be edited to reflect that the intervention was not in fact ‘Short term aerobic exercise’; but was ‘Low intensity aerobic exercise’
Other issues:
There is Responder group analysis but it was not clear to me how ‘responders’ were identified. The term ‘responder’ suggest that the group improved in the primary outcome but in Table 7 there were no significant changes in the cognitive outcomes. In order to better understand responder outcomes it would help if there was further analysis. For example, was there a significant relationship between attendance and change in memory score (post-pre)? Or a relationship between improvement in fitness and cognitive score change?
There was 12 week follow-up but I could not find any mention of the results of the follow-up.
Where are the F values and effect sizes for the ANCOVAs?

---

## Round 0.2 · accepted · Accept

I have now received all the reviewers’ comment with satisfaction of your reply and revisions from previous comments. You and your coauthors have my congratulations. Thank you for choosing PeerJ as a venue for publishing your research work and I look forward to receiving more of your work in the future.

Tsung-Min Hung, PhD., FNAK
PeerJ editor
Research chair professor,
Department of Physical Education,
National Taiwan Normal University

# Reviewer 1 ·

Basic reporting

no comment

Experimental design

no comment

Validity of the findings

no comment

Additional comments

I am satisfied with all the revisions to the manuscript and have no further comment. The authors should be credited by their inputs to the manuscript.

·

Basic reporting

The authors have adequately addressed my questions.

Experimental design

Design and study elements are now clear.

Validity of the findings

Valid and important findings.